# Study on the Synergistic Molluscicidal Effect of Pedunsaponin A and Niclosamide

**DOI:** 10.3390/molecules27217623

**Published:** 2022-11-07

**Authors:** Chunping Yang, Yue Zhou, Chuanlei Wu, Xiao Yan, Pengcheng Cheng, Liya Luo, Xiaoyan Qiu, Min Zhang, Guangwei Qin, Yangyang Zhang, Huabao Chen

**Affiliations:** 1College of Agronomy, Sichuan Agricultural University, Chengdu 611130, China; 2Shaoxing Customs, Shaoxing 312000, China

**Keywords:** pedunsaponin A, Niclosamide, *Pomacea canaliculate*, synergistic

## Abstract

Niclosamide (NI) is the main molluscicide used to control *Pomacea canaliculata* (Lamarck) (Architaenioglossa: Ampullariidae). However, NI failed to inhibit snail climbing during the treatment process. In this study, we examined the effect of NI combined with pedunsaponin A at an ineffective concentration. The molluscicidal effect of Pedunsaponin A on NI was evidently synergistic after 48 h, and the synergism ratio (SR) was 1.82 after treatment for 72 h at 0.8 mg·L^−1^. Examination of the climbing adhesion effect showed that a high concentration of Pedunsaponin A (0.4 mg·L^−1^ and 0.8 mg·L^−1^) combined with NI significantly inhibited the climbing of *P. canaliculata*. We further studied the synergism mechanism; the results of histopathological observation showed that the siphon appeared cavities, the muscle fibers of the ventricular were severely dissolved, and kidney tubule arrangement was distorted after NI adding Pedunsaponin A. In addition, the hemocyte survival rate and the content of hemocyanin decreased significantly. According to the results of our study, the synergism mechanism may hinder oxygen transport of *P. canaliculata*, influencing the supply of energy; the ability of immune defense and excretion and metabolic detoxification decreased, prolonging the action time of NI in the body.

## 1. Introduction

The *Pomacea canaliculata* (Lamarck) (Architaenioglossa: Ampullariidae) is a mollusk and is also known as the apple snail [1]. Because of its wide range of food, fast growth, strong fecundity, and adaptability, it is classified as invasive species. Apple snail causes huge rice yield loss and is reported frequently around the world [2]. Additionally, the apple snail is also an intermediate host of *Angiostrongylus cantonensis*, which has caused a series of health problems in people [3]. Therefore, controlling this harmful snail not only helps to reduce crop yield loss but also protects people from *Angiostrongylus cantonensis* disease infection.

At present, NI is the main method used to control apple snails, but during the treatment process, we found that NI failed to inhibit snail climbing, and thus its efficacy was reduced. Previous research found that some plant extracts could significantly inhibit snail climbing and exhibit high molluscicidal activity after being combined with NI. The authors of [4] found that a high concentration of arecoline can inhibit muscle contraction and climbing of the snail, thus prolonging the action time of NI against the snail [5] found that an alkaloid of *Eomecon chionantha Hance* has a synergistic effect with NI, increasing mortality in snails. The authors of [6] also found that *Alternanthera philoxeroides (Mart.) Griseb* could inhibit *Oncomelania hupensis* (Gastropoda: Rissooidea) climbing and escape from the water, presenting a synergistic effect with NI.

*Pueraria peduncularis* (Grah. ex Benth.) Benth is a flowering herb that is widely distributed throughout southwestern China. The Biorational Pesticide Laboratory of Sichuan Agricultural University found that the extract of *P. peduncularis* could significantly inhibit snail climbing and presented strong molluscicidal activity against *P. canaliculate* [7]. After the separation and identification of the main active substances in the extract, its chemical structure was obtained and named Pedunsaponin A (Figure 1). A previous study found that Pedunsaponin A had significant toxic effects on different organs of the snail, especially the cilia of lungs and gills; subsequently, Pedunsaponin A into the hemolymph and destroys the hemocyte morphology, resulting in the destruction of the immune system in the snail. It is mainly manifested as aggravation of the depolarization phenomenon of the blood cell membrane and death rate. With the increase in the cell number, the deformation of nuclei and cytoplasm will intensify, and part of the cytoplasm will flow out in a large amount, causing the nuclei to disintegrate or even escape from the cells. Further study found that PcnWAS is one of the important target proteins of PA in *P. canaliculata*. and that advillin (PcAdv) in the gills of *P. canaliculata* is the key target protein. In this study, we systematically study the synergistic molluscicidal effect of Pedunsaponin A combined with NI and lay a foundation for the development and application of Pedunsaponin A.

## 2. Results

### 2.1. Ineffectual Snail-Killing Concentration of Pedunsaponin A

As shown in Table 1, the immersion toxicity effects of different concentrations of Pedunsaponin A on *P. canaliculata* were significantly different. Snail mortality was extremely low at Pedunsaponin A concentrations of 0.2, 0.4, and 0.8 mg·L^−1^; therefore, these concentrations were selected as the ineffectual killing snail concentrations, and a further study of the synergistic effect of Pedunsaponin A combined with NI at different ineffectual concentrations was performed.

### 2.2. The Synergistic Effect of Pedunsaponin A on NI

The immersion toxicity results of different NI to *P. canaliculata* are shown in Table 2. The corrected mortality of *P. canaliculata* was raised as the concentration of NI increased. Additionally, as the treatment time of NI to *P. canaliculata* prolonged and the corrected mortality also increased. The LC_50_ values at 24 h, 48 h, and 72 h were 1.0431 mg·L^−1^, 0.9034 mg·L^−1^, and 0.8999 mg·L^−1^, respectively.

The synergistic effects of different concentrations of Pedunsaponin A on NI are shown in Table 3. At 24 h, 48 h, and 72 h, the synergism ratios of NI at 0.2 mg·L^−1^ Pedunsaponin A were 0.89, 1.2, and 1.33, respectively; the synergism ratios of NI at 0.4 mg·L^−1^ Pedunsaponin A were 0.86, 1.41, and 1.53, respectively; and the synergism ratios of NI at 0.8 mg·L^−1^ Pedunsaponin A were 0.40, 1.48, and 1.82, respectively. The data demonstrate that the molluscicidal effect of Pedunsaponin A on NI is antagonistic in 24 h, and with increasing concentrations, the antagonism becomes obvious. However, after 48 h, Pedunsaponin A shows significant synergism with NI, and the synergistic effect is proportional to the concentration and time.

### 2.3. The Climbing Adhesion Effect of Pedunsaponin A on NI

The climbing adhesion rates of *P. canaliculata* within 24 h after the application of combinations of different concentrations of Pedunsaponin A and NI are presented in Figure 2. There was no significant difference in the climbing adhesion rate observed under the NI and 0.2 mg·L^−1^ Pedunsaponin A treatment. However, when combined with 0.4 mg·L^−1^ or 0.8 mg·L^−1^ of Pedunsaponin A, the climbing adhesion rate significantly decreased. The above results proved that the addition of Pedunsaponin A can solve the problem of NI being ineffective in inhibiting snail climbing.

### 2.4. Oxygen Consumption Effect of Pedunsaponin A on NI

The effects of different concentrations of Pedunsaponin A combined with NI on the oxygen consumption of *P. canaliculata* are shown in Figure 3. It can be seen from the figure that at 12 h, NI or the combination of the two significantly reduced the oxygen consumption of snails and inhibited their respiration, but the difference between treatments was not significant. With the prolongation of time, adding Pedunsaponin A to NI could significantly reduce the oxygen consumption and inhibit their respiration compared with alone NI at 24 h. However, the difference between each treatment was not significant at 48 h.

### 2.5. Histopathological Changes in Various Organs

The histopathological changes in the four organs were examined after 48 h. According to the results shown in Figure 4(A1–A4), NI did not produce an obvious influence on the siphon (A3) compared with CK (A1), but after MI treatment, there were many cavities in the siphon.

According to the results shown in Figure 4(B1–B4), the morphology of the ventricle between NI (B3) and MI (B4) was significantly different compared with CK (B1). The normal ventricle of the snail has plenty of muscle fibers that are tightly packed and clearly visible (B1). The muscle fibers in NI were lightly dissolved, and lacuna were exhibited. While the muscle fibers in MI (B4) were severely dissolved and dissociated, and larger lacuna appeared.

According to the results shown in Figure 4(C1–C4), compared with NI (C1), the lacuna enlarged in MI and showed the glandular parenchyma cells became loosening and rearrangement after MI treatment (C4).

According to the results shown in Figure 4(D1–D4), compared with CK (D1), except for a few separate kidney tubules, most of the tubules were arranged neatly (D3) after treatment with NI, while the arrangement of kidney tubules was disordered and cavities emerged after treatment with MI (D4), indicates the addition of Pedunsaponin A aggravates kidney damage.

### 2.6. The Effect of Different Treatments on the Hemocyanin Content in Hemocytes

According to the results shown in Figure 5, at 48 h, the difference in hemocyanin content between Pedunsaponin A, NI, and MI was not significant compared with that of CK. At 60 h and 72 h, the content of hemocyanin in Pedunsaponin A was similar to MI but significantly lower than that with NI. This result indicated that Pedunsaponin A caused a decrease in hemocyanin content compared with NI alone.

### 2.7. The Effect of Different Treatments on P. canaliculata Hemocytes

#### 2.7.1. The Effect of Different Treatments on the Survival Rate of Hemocytes

According to the results shown in Figure 6, the hemocyte survival rate in MI was similar to that in Pedunsaponin A and significantly lower than that in NI at 48 h and 60 h. The hemocyte survival rate in MI was similar to that in Pedunsaponin A and lower than that in NI at 72 h, but the difference was not significant. This result indicated that Pedunsaponin A combined with NI increased the mortality of the hemocytes.

#### 2.7.2. The Effect of Different Treatments on the Morphology of Basophilic Granulocytes

According to the results shown in Figure 7, the normal basophilic granulocytes were usually oval in shape, the nuclei were pale violet red, and the cytoplasm was light blue after staining with the Wright–Giemsa stain (A1). At 48 h, the cytoplasm was slightly deformed after treatment with NI and MI (A3, A4), but there was no difference. At 60 h, the cytoplasm was enlarged in MI (B4); at 72 h, the cytoplasm in NI (C3) was not different from that in CK (C1), while it was solid and tightly packaged around the nucleus after treatment with MI (C4). The results show the mixture of Pedunsaponin A and NI enhanced the damaging effects on the morphology of basophilic granulocytes.

## 3. Materials and Methods

### 3.1. Animals

In this study, according to the shell height (h) of *Pomacea canaliculata* (Lamarck) (Architaenioglossa: Ampullariidae), adult female *P. canaliculata* (25 mm ≤ H < 40 mm) was selected for the experiment [8], which was collected from Chengdu City (30.71° N, 103.87° E), Sichuan Province in June 2018. The snails were continuously cultured for at least 5 d to acclimatize to the laboratory conditions. The apple snails were placed in 50 L plastic aquariums and were fed with fresh cabbage leaves. Water and dead snails were removed daily from these aquariums. Feeding of all snails was carried out at room temperature with natural light conditions and without artificial aeration. The weight of the snails used in this study was 5 ± 0.5 g.

### 3.2. Preparation of Pedunsaponin A and NI

Roots of *Pueraria peduncularis* (Grah. ex Benth.) Benth was collected from wild populations in Yaan City (29.90° N, 102.92° E), Sichuan Province, in July. Pedunsaponin A was isolated from the root extracts of *P. peduncularis* by the Biorational Pesticide Laboratory of Sichuan Agricultural University [7]. The purity of Pedunsaponin A was greater than 98%. NI (70%) was purchased from Zigong Hengda Pesticide Company, Sichuan Province, Zigong, China.

### 3.3. Screening of Ineffectual Snail-Killing Concentrations of Pedunsaponin A

Pedunsaponin A was prepared in the mother liquor of 1000 mg·L^−1^, which was used to generate solutions containing Pedunsaponin A at 6.4, 3.2, 1.6, 0.8, 0.4, and 0.2 mg·L^−1^. Thirty snails were placed in each 1 L glass beaker, which was covered with a clean nylon net to prevent escape, and incubated at 25 °C in a thermostat-controlled water bat; distilled water was used as a control, and three replicates were conducted for the experiment. The mortality was recorded after 24 h, 48 h, and 72 h; the corrected mortality percentages were assessed by Abbott’s formula, and all data were analyzed using DPS 9.50. The highest concentration of Pedunsaponin A was an ineffectual snail-skilling concentration under the condition of without *P. canaliculata* dead.

### 3.4. The Synergistic Effect of Pedunsaponin A Combined with NI

#### 3.4.1. Determination of the LC_50_ of NI

Assessment of toxicity of NI on *P. canaliculata* was conducted by the immersion method. In each bioassay, five serial concentrations with three replications, including control, were performed, in which the concentrations of NI were 0.7, 0.8, 0.9, 1.0, and 1.1 mg·L^−1^, respectively. The mortality was recorded at 24 h, 48 h, 72 h, LC_50_, and 95% confidence interval and calculated according to the probability analytical method and least square method [9]. Experimental treatments method were the same as 2.3.

#### 3.4.2. Determination of Synergism Ratio of Pedunsaponin A and NI

Three treatments were established, 0.2, 0.4, and 0.8 mg·L^−1^ Pedunsaponin A was added to five serial concentrations of NI, respectively, according to the assessment of toxicity of NI on *P. canaliculata*. Three replications were set up; distilled water was used to bank control. Mortality was recorded at 24 h, 48 h, and 72 h. Subsequently, the regression equation, LC_50_, and synergism ratio were calculated; the synergism ratios are calculated as follows:(1)$$\mathrm{Synergism}\;\mathrm{ratio}\;\left(\mathrm{SR}\right)=\frac{{\mathrm{agent}\;\mathrm{LC}}_{50}}{\left(\mathrm{agent}+{\mathrm{synergistic}\;\mathrm{agent}\;\mathrm{LC}}_{50}\right)}$$

SR > 1 indicates a synergistic effect.SR = 1 indicates no synergistic effect.SR < 1 indicates antagonism.

#### 3.4.3. Determination of the Climbing Adhesion Effect of Pedunsaponin A on NI

Adding 0.2, 0.4, and 0.8 mg·L^−1^ Pedunsaponin A to 0.7 mg·L^−1^ NI solution, three treatments were established, and distilled water was used as bank control. Observe the condition in which the snail climbs from the liquid within 24 h, based on which the climbing adhesion rate was assessed.

### 3.5. Determination of the Oxygen Consumption Rate

In these tests, three concentrations of Pedunsaponin A (0.2, 0.4, and 0.8 mg·L^−1^) were combined with 0.6 mg·L^−1^ NI. The changes in the oxygen consumption rate (R_0_) of the snails under different treatments were determined.

Three snails of the same size were placed in a 500 mL iodine flask for 12 h, 24 h, or 48 h. After completion of the treatment, the water in the flask was used to determine the dissolved oxygen content. Dissolved oxygen was determined via Winkler iodometry. In this test, distilled water was used as the blank control, and Pedunsaponin A and NI were added separately as drug controls; each treatment was repeated three times.

R_0_ was calculated as follows:R_0_ = (O_0_ − O_t_) · V/(t · n · W)(2)

In these equations, O_0_ and O_t_ are dissolved oxygen concentrations (mg·L^−1^) in the blank control and experimental groups, respectively, at the end of the experiment. All data were analyzed with DPS 9.50, means data were compared using Duncan’s multiple range test at *p* ≤ 0.05.

t: the time for processing a snail.V: the volume of water used in the experiment.n: the number of experimental snails.W: the dry weight of the soft tissue.

### 3.6. The Effect of Different Treatments on the Morphology of Various Organs in P. canaliculata

Four treatment groups were established: 0.8 mg·L^−1^ Pedunsaponin A solution, 0.6 mg·L^−1^ NI solution, MI solution (0.8 mg·L^−1^ Pedunsaponin A and 0.6 mg·L^−1^, MI mixture), and the control group (CK, water), each treatment was repeated three times. Each replicate was thirty snails and treated within 48 h. Following treatment, both the treated and control snails were removed from their shells for extraction of their organs, including the siphon, ventricle, pericardial, and kidney. The histopathological changes were observed under a light microscope by the paraffin section method.

### 3.7. The Effect of Different Treatments on the Content of P. canaliculata Hemocyanin

The treatment groups were the same as described in 2.6. The shell around the heart of *P. canaliculata* was removed by dissecting scissors, and a 1 mL disposable syringe was used to suction hemocytes into a 1.5 mL centrifuge tube preloaded with anticoagulant solution (1% heparin sodium, Beijing Solarbio Science & Technology Co., Ltd., Beijing, China). The supernatant was retained after centrifugation for 3 min (1000 rpm). The UV spectrophotometry method was adopted to estimate the hemocyanin content of hemocytes. The supernatant was made into a 1% diluent and measured on a spectrophotometer (334 nm). The hemocyanin content was 2.69 times the optical density value under a light diameter of 1 cm. All data were analyzed with DPS 9.5. The formula for the calculation of the hemocyanin content was as follows:(3)$$\mathrm{Hemocyanin}\;\mathrm{content}=2.69E_{1\mathrm{cm}}^{1\%}\;\mathrm{nmol}/L$$

### 3.8. The Effects of Different Treatments on the Hemocytes

#### 3.8.1. Measurement of the Survival Rate of Hemocytes

The trypan blue dye method was used as described by [10], the treatment groups were the same as mentioned in Section 2.6, and 1 mL disposable syringe was used to suction hemocytes into a 1.5 mL centrifuge tube preloaded with anticoagulant solution. The sediment was retained after centrifugation for 5 min, and the supernatant was discarded. PBS solution (pH 7.2–7.4, Bejing Solarbio Science & Technology Co., Ltd., Beijing, China) was added to wash the deposits; after centrifugation and surpernation were discarded, the sediment was retained, and this step was repeated once more. Subsequently, the hemocyte deposits were then re-suspended. The cell density was adjusted to 1 × 10^6^ cells·mL^−1^, and a 45 µL hemocyte suspension was mixed with 5 µL of 0.4% trypan blue solution (Beijing Solarbio Science & Technology Co., Ltd., Beijing, China). Within 3 min, a blood cell counting plate board was used to count live and dead cells under a light microscope. The dead cells were stained blue, and the live cells were colorless and transparent. Three independent replicates were conducted, all data were analyzed with DPS 9.50, and the hemocytes survival rate was calculated by the following formula:(4)$$\mathrm{Survival}\;\mathrm{rate}\left(\%\right)=\frac{\;\mathrm{total}\;\mathrm{number}\;\mathrm{of}\;\mathrm{living}\;\mathrm{cells}}{\mathrm{total}\;\mathrm{number}\;\mathrm{of}\;\mathrm{living}\;\mathrm{cells}+\mathrm{total}\;\mathrm{number}\;\mathrm{of}\;\mathrm{dead}\;\mathrm{cells}}\times 100$$

#### 3.8.2. The Effect of Different Treatments on the Morphology of Basophilic Granulocytes

The treatment groups were the same as mentioned in 2.6, and the method of hemocyte extraction was the same as mentioned in 2.7. A total of 20 µL of hemocytes was dropped onto a glass slide to make a blood smear and then dried at room temperature. Subsequently, the hemocyte smears were fixed with methanol for 4 min and immersed in Wright–Giemsa stain for 5 min before being washed in PBS and air-dried. Then, the dried blood smears were sealed with a neutral gum seal and dried in an oven at 38 °C for 48 h [11]. Basophilic granulocytes were examined and counted under a light microscope. Three independent replicates were conducted, and all data were analyzed with DPS 9.50.

## 4. Conclusions and Discussion

*Pomacea canaliculata* (Lamarck) (Architaenioglossa: Ampullariidae) has rapidly spread and become an important invasive species since it was introduced to China. NI is a unique one for recommending molluscicide by WHO, which exhibits good molluscicidal effects and is safe for humans and livestock [12]. However, *P. canaliculata* is easy to climb from water under effective concentration, influencing molluscicidal effect. In recent years, many plant extracts combined with NI presented synergistic effects, for instance, the plant extract of *Glycyrrhiza uralensis* Fisch [13], alkaloids from *Eomecon chionantha* Hance [5], and *Eucalyptus camaldulensis* Dehnh extract [14]. In this study, we found that after Pedunsaponin A was combined with NI, the climbing escape phenomenon of snails reduced significantly, and the synergistic effect was significant after 48 h. It can be seen that Pedunsaponin A combined with NI can effectively make up for the disadvantages of NI, enhanced molluscicidal effect, and has broad application prospects.

*Pomacea canaliculata* (Lamarck) (Architaenioglossa: Ampullariidae) carried out the oxygen exchange mainly by pectinate gill [15,16]. The exchange of oxygen is transported to various organs and tissues through blood circulation, completed the oxidation metabolism provides energy for the body. The ventricular, pericardial cavity, and auricle play an important role [17]. The main function of the ventricular is to pump arterial blood out of the main artery; pericardial cavity receives jugular venous from the gills and then sends it to the auricle for circulation. The experiment showed that after adding Pedunsaponin A to NI, the content of hemocyanin decreased significantly, the respiratory oxygen consumption rate reduced, the siphon appeared cavities, and the muscle fibers of the ventricular were severely dissolved and dissociated. Based on the above results, we speculated that Pedunsaponin A could destroy oxygen transported by *P. canaliculata*, influencing energy supply is an important reason for the increasing molluscicidal effect of NI.

The immune defense process of *P. canaliculata* is mainly coordinated by hemocytes involved with cellular immune and humoral immunity mediated by related immune enzymes [18,19,20]. The hemocyte has complex and complete defensive functions, which can clear intracorporal foreign bodies by engulfing, encapsulation, storage, and bacteriolysis [21]; the immune response mechanism will be disturbed when the hemocyte is damaged. [22] found that Pedunsaponin A can destroy hemocytes and lead to the death of *P. canaliculata*. This study found that Pedunsaponin A combined with NI could destroy the morphology of basophilic granulocytes and decrease the hemocyte survival rate, which indicated that the immune system of *P. canaliculata* was influenced; this might be another important factor for Pedunsaponin A to increase the molluscicidal effect of NI.

The kidney of *P. canaliculata* is an important organ for the accumulation, detoxification, and excretion of toxicants. The potential pathway of *P. canaliculata* to discharge waste metabolites is via immersion in blood sinus tubular epithelial cells, and the waste metabolites are transferred to the lumen by tubular epithelial cells and eliminated after reabsorption [23]. In this study, kidney tubule arrangement became more distorted, and organizational cavity appeared after treatment of Pedunsaponin A combined with NI. These results showed that adding Pedunsaponin A could lead to the abnormal physiological function of the kidney in snails, influencing the ability of excretory and detoxification, which might be the third factor in synergistic effects.

## Figures and Tables

**Figure 1 molecules-27-07623-f001:**
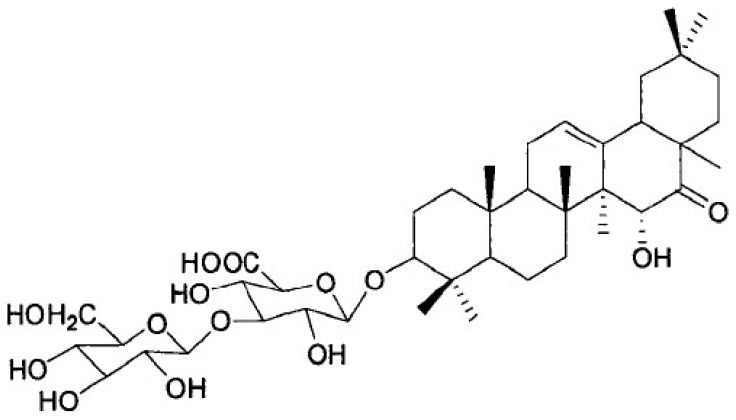
The chemical structure of Pedunsaponin A.

**Figure 2 molecules-27-07623-f002:**
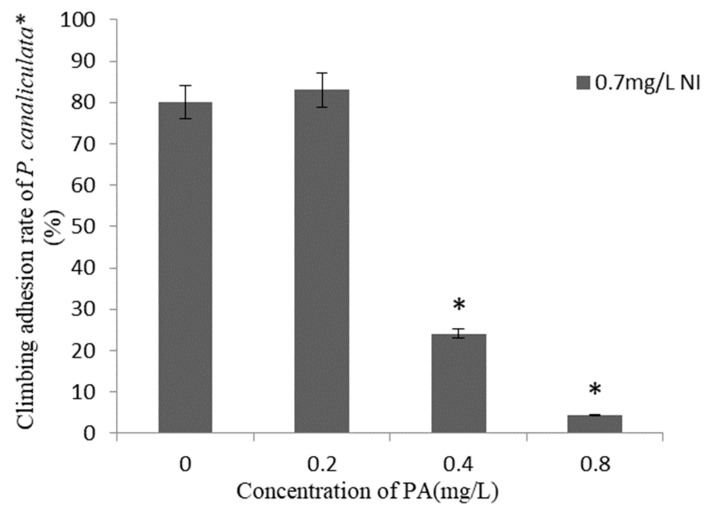
Climbing adhesion rate of *P. canaliculata* under treatment with Pedunsaponin A on NI for 24 h, *—statistically significant.

**Figure 3 molecules-27-07623-f003:**
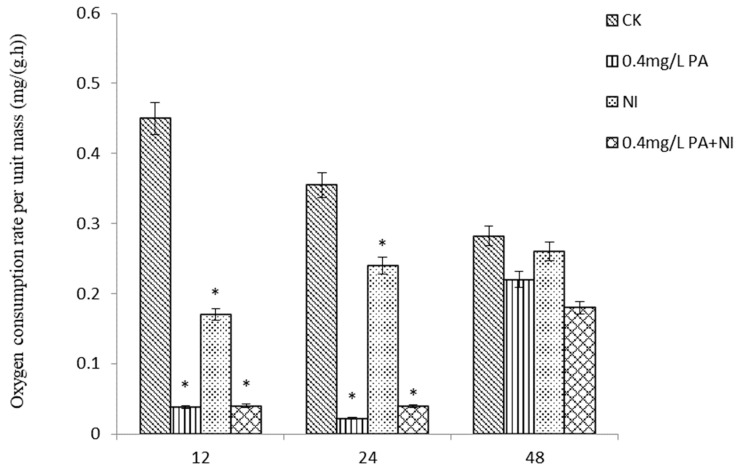
The effect of the drug on the oxygen consumption rate. *—*p* < 0.05.

**Figure 4 molecules-27-07623-f004:**
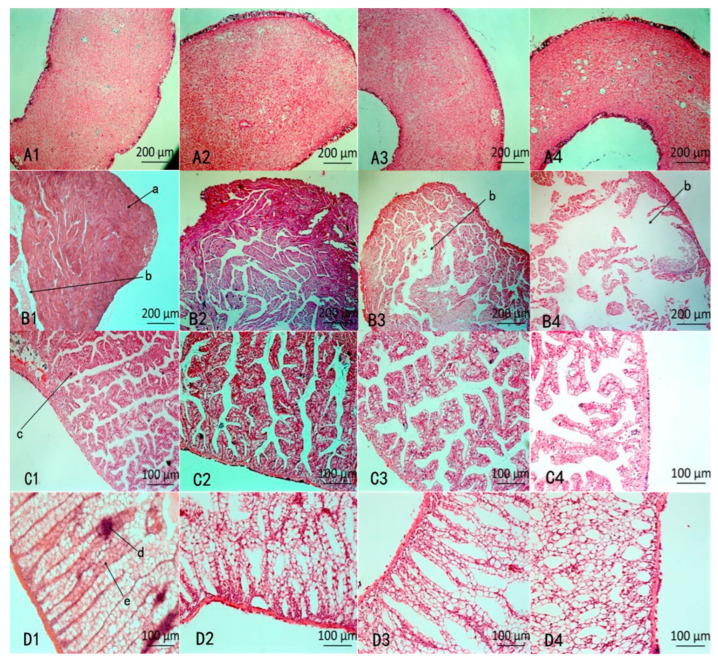
Histopathological changes in various organs. (**A1**–**D1**) represent CK (water treatment), (**A2**–**D2**) represent PA treatment, (**A3**–**D3**) represent NI treatment, (**A4**–**D4**) represent MI treatment; A: Siphon; B: Ventricle; C: Pericardial gland; D: Kidney; a: muscle fiber; b: gap; c: parenchyma cell; d: collecting tube e: kidney tubules.

**Figure 5 molecules-27-07623-f005:**
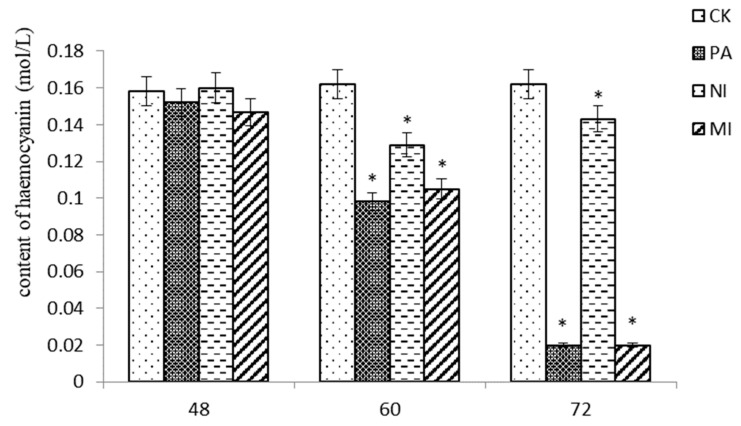
The effect of different treatments on the hemocyanin content in hemocytes, *—*p* < 0.05.

**Figure 6 molecules-27-07623-f006:**
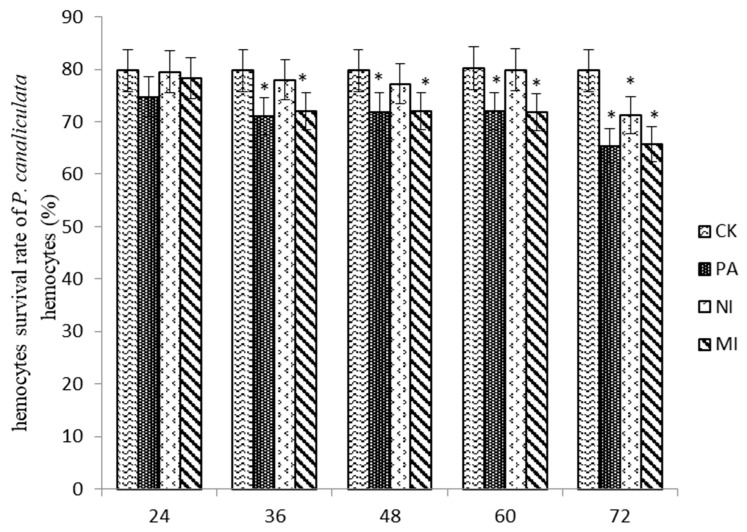
The effect of different treatments on the survival rate of *P. canaliculata* hemocytes, *—*p* < 0.05.

**Figure 7 molecules-27-07623-f007:**
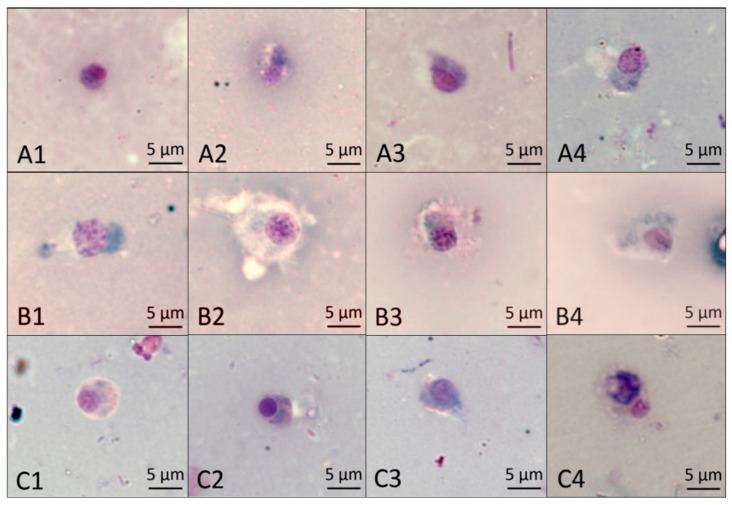
The effect of different treatments on *P. canaliculata* hemocyte basophilic granulocytes. (**A1**–**A4**) show the effect of different treatments, CK, PA, NI, and MI, on basophilic granulocytes at 48 h; (**B1**–**B4**) show the effect of different treatments, CK, PA, NI, and MI, on basophilic granulocytes at 60 h; (**C1**–**C4**) show the effect of different treatments, CK, PA, NI, and MI, on basophilic granulocytes at 72 h.

**Table 1 molecules-27-07623-t001:** Immersion toxicity effect of different concentrations of Pedunsaponin A on *P. canaliculata*.

Concentration (mg/L)	Corrected Mortality (%)		
	24 h	48 h	72 h
0.2	0 d	0 d	0 d
0.4	0 d	0 d	0 d
0.8	0 d	0 d	0 d
1.6	9.33 c	12.67 c	15.67 c
3.2	16.33 b	31.33 b	47.13 b
6.4	30.00 a	56.00 a	89.28 a

Note: The data followed by the same small letter in the column are not significant difference at the 0.05 probability level.

**Table 2 molecules-27-07623-t002:** Immersion toxicity effect of different concentrations of NI for *P. canaliculata*.

Treatment Concentration	Corrected Mortality (%)		
	24 h	48 h	72 h
0.7	13.33 e	16.67 e	16.67 e
0.8	23.33 d	23.33 d	26.67 d
0.9	30.00 c	46.67 c	46.67 c
1.0	46.67 b	70.00 b	70.00 b
1.1	56.67 a	90.00 a	90.00 a
LC_50_	1.0431	0.9034	0.8999
Correlation coefficient	0.9933	0.9755	0.9845
95% confidence interval	0.9428~1.1540	0.8533~0.9349	0.8493~0.9317

Note: The data followed by the same small letter in the column are not significant difference at the 0.05 probability level.

**Table 3 molecules-27-07623-t003:** Synergistic effects of different concentrations of Pedunsaponin A on NI for *P. canaliculata*.

Time (h)	Added Concentration (mg/L)	Regression Equation	LC_50_ (mg/L)	Correlation Coefficient	Synergism Ratio	Slope ± SE
24 h	0.2	y = 4.5925 + 5.9524x	1.17	0.97	0.89	5.9524 ± 0.0209
	0.4	y = 4.4011 + 6.8895x	1.22	0.97	0.85	6.8895 ± 0.0212
	0.8	y = 3.0887 + 4.5915x	2.61	0.98	0.40	4.5915 ± 0.0550
48 h	0.2	y = 6.5413 + 11.7774x	0.74	0.99	1.20	11.7774 ± 0.0079
	0.4	y = 6.4724 + 7.3753x	0.63	0.95	1.41	7.3753 ± 0.0224
	0.8	y = 6.7180 + 8.0016x	0.61	0.95	1.46	8.0016 ± 0.0135
72 h	0.2	y = 6.6245 + 9.0568x	0.66	0.97	1.33	9.0568 ± 0.0158
	0.4	y = 6.6734 + 7.0745x	0.58	0.97	1.53	7.0745 ± 0.0306
	0.8	y = 5.8028 + 2.6054x	0.49	0.99	1.82	2.6054 ± 0.0283

## Data Availability

Not applicable.

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
