# Peer review of "Study on the Synergistic Molluscicidal Effect of Pedunsaponin A and Niclosamide"

_molecules, 2022, doi:10.3390/molecules27217623_

Round 1

Reviewer 1 Report

Yang and Zhou studied the synergistic effect of Pedusaponin A and Niclosamide. However, the introduction and the whole paper have a limited bibliografy that must be impruved. Moreover, all data lack of standard deviation (e.g., table 1, table 3) and graphs lack of any error bars (e.g., Figure 2 and 3). Figure should all be aligned (Figure 4) and show similar colours to be compered (Figure 7) . The age and the sex ratio of animals used during the experiments are not reported. Other experiments should be provided to clarify the sinergistic effect of pedsaponin at the molecular level. In my opinion, the whole paper must be totally rethought.

Author Response

请看附件

Reviewer 2 Report

Point-by-Point

Molecules – Manuscript # 1985271

Dear Editor and Author(s),

The study explores the potential synergism between Niclosamide (i.e., molluscicide) and pedunsaponin A (i.e., a substance of botanical origin), including toxicological, histopathological, and physiological outcomes. The research results were interesting by providing a promising source for pest management. However, I have a few suggestions in order to improve the quality of the manuscript before publication.

Overall comments.

The introduction highlights the main protagonists in the text and exhibits the potential importance of research. The argument is robust and was based on the potential synergism action of Niclosamide (i.e., molluscicide) and pedunsaponin A. The authors did not, however, specify the reasons for and aims for toxicological, histological, and physiological results. They simply state, “In this study, we systematically study the synergistic molluscicidal effect of Pedunsaponin A combined with NI and lay a foundation for the development and application of Pedunsaponin A.” (see Lines 51-53), which appears to be a general purpose with little details. Therefore, I recommend careful revision in order to better address the readers and avoid misunderstanding.

The Material and methods are detailed and easy to understand. However, the statistical techniques were not precise. Are the assumptions of normality and homoscedasticity verified? Are the mortality results from 24h, 48h, and 72h cumulative? My primary point of worry is with toxicological data, which is based on percentages and perhaps not normally distributed. I strongly recommend a detailed revision on this point.

The Results and Discussion are remarkably interesting and easy to understand. I merely feel a little confused by the histology and physiology bioassay methods and results since there is no appropriate context in the introduction and aims. Even so, I think that they are crucial for interpreting the results and unquestionably raise the paper's quality.

I hope that the comments improve your manuscript,

All the best

Reviewer.

Round 2

Reviewer 1 Report

Thank you for your corrections. However, I have other suggestions:

1.      Histograms lack of any symbols (e.g., *) to highlight statistically significant results and you need to insert the p-value in the caption of the figure. This suggestion is strongly recommended also for tables.

2.      Figure 7. B3 – The image reported into the attached pdf I see a completely different image compared to other ones. Moreover, it seems overlap another image.

At least, I suggest to improve your introduction and reference sections considering also works where other molecules or strategy where used to prevent P. canaliculata spread.
